# Primary postpartum haemorrhage at the Libreville University Hospital Centre: Epidemiological profile of women

Nathalie Ledaga Ambounda[1], Sylvain Honore Woromogo [2]*, Felicite-Emma Yagata-Moussa[3], Liela Agnes Okoyi Ossouka[4], Vicky Noel Simo Tekem[5], Eliane Okira Ango[6], Alain Jepang Kouanang[1]

**1** University Hospital Centre of Libreville, Gynecology and Obstetric Service, Liberville, Gabon, **2** InterStates Centre for Higher Public Health Education in Central Africa (CIESPAC), Brazzaville, Congo, **3** Faculty of Health Sciences, University of Bangui, Bangui, Central African Republic, **4** Army Training Hospital Omar Mbongo Ondimba, Libreville, Gabon, **5** National Cancer Prevention and Control Programme, Ministry of Public Health, Libreville, Gabon, **6** Health Centre Nzeng Ayong, Libreville, Gabon

* woromogos@gmail.com

**Data Availability Statement:** All relevant data are within the paper.

## Abstract

In Gabon, the proportion of maternal deaths directly related to Primary PostPartum Haemorrhage (PPPH) is 15 to 25%, despite the different means that the World Health Organization has made available to the providers of Emergency Obstetrical and Neonatal Care (EmONC). The objective of this study was to determine the prevalence and epidemiological characteristics of Primary PostPartum Haemorrhage to improve its management and reduce the rate of maternal deaths. An analytical retrospective study involved 42,728 records, whose data were collected using a chart collection form on the basis of information contained in partograms and other patient records. Sociodemographic variables were expressed using percentage. The relationship between the etiologies of PPPH and certain characteristics of the women was established using the ORs with their 95% confidence intervals. The difference was significant if p < 0.05. The prevalence of PPPH was 1.6%. Delivery haemorrhages accounted for 65.5% of PPPH. The main factors associated with delivery haemorrhages were pauci parity and multiparity (p = 0.003 and 0.051), post-term (p = 0.042), and birth weight >4,000 g (p = 0.006). Those associated with genital tract injuries were young maternal age (p = 0.008) and multiparity (p = 0.028). The most common etiology was haemorrhage from delivery. Multiparity remains the most common risk factor and the young age of the patients. It is important to improve management through better assessment of blood loss in the primary postpartum period as well as capacity building of health providers on EmONC.

## Introduction

Primary postpartum hemorrhage (PPPH) is defined as the loss of 500 mL or more of blood within 24 hours of delivery [1–3]. In developing countries and sub-Saharan Africa, it is the

**Funding:** The authors received no specific funding for this work.

**Competing interests:** The authors have declared that no competing interests exist.

leading direct cause of maternal death [2,4–7]. In Gabon, the frequency of maternal deaths directly related to PPPH is clearly increasing at the Libreville University Hospital Centre. It increased from 15% to 25% between 2013 and 2015 [8], despite the various means that World Health Organization (WHO) has made available to Emergency Obstetrical and Neonatal Care (EmONC) providers, such as the implementation of recommended protocols on the management of postpartum haemorrhage. Some of these deaths were preventable [9–11]. The objective of this study was to determine the prevalence and epidemiological characteristics of PPPH at Libreville University Hospital Centre with the aim of improving its management and reducing the rate of maternal deaths.

## Methods

### Purpose and type of study

The study took place in the Department of Gynaecology and Obstetrics of the University Hospital Centre of Libreville. This department has a maternity ward with a capacity of 60 patients and a delivery room with 13 delivery beds. This was an analytical retrospective study over a 5-year period from January 1, 2010, to December 31, 2014, based on the systematic collection of primary postpartum haemorrhage case files that occurred during this period.

### Study population and sampling

The study population is represented by all parturients admitted to the Department of Gynaecology and Obstetrics during the study period, N = 42728. Included were all vaginal deliveries and cesarean deliveries with primary postpartum hemorrhage. Not included were deliveries with secondary postpartum hemorrhage, postabortion hemorrhage, and rectorrhage or hematuria. All records of parturients admitted to the Gynaecology and Obstetrics Department of the University Hospital Centre of Libreville who had presented an primary postpartum haemorrhage and who were registered were collected. The data were collected using a collection form. The form was filled in on the basis of information contained in partogrammes, birth registers, surgical reports, anaesthesia and reanimation registers and hospitalization registers.

### The variables

The variables studied were sociodemographic characteristics, obstetric and gynaecological history of women, characteristics of the pregnancy, the conduct of delivery and the etiologies of haemorrhage.

Spontaneous delivery is when the placenta is expelled under the effects of abdominal thrust with the patient in an upright position. Directed delivery is based on prophylactic administration of synthetic oxytocin (5–10 IU), accompanied by continuous traction on the cord and upward displacement of the uterine body. Artificial delivery is assisted by the caregiver with certain techniques.

### Data analysis

The data were entered into an Excel database and analyzed using SPSS version 22. We categorised these haemorrhages into two groups. The group of haemorrhages due to placental anomalies including uterine atony and the group of haemorrhages of traumatic causes of the genital tract. Sociodemographic variables were expressed using percentage. The relationship between the etiologies of PPPH and certain characteristics of the women was established using simple and multiple logistic regression, including potential confounding factors. For this purpose, the Chi-square and Wald tests were used as well as the odds ratio with their 95% confidence. The

difference was significant if p < 0.05. Multivariable logistic regression analyses with backward elimination stepwise selection with p < 0.20 were used to identify baseline explication that predicts PPPH.

### Ethical considerations

The study received the approval of the ethics committee and an authorization from the Direction of the University Hospital of Libreville. Authority to conduct the research was sought from the University of Health Sciences of Gabon. Ethical clearance to conduct the study was sought from the Ethical Review Committee of the University of Health Sciences with reference number 916/USS/FMSS/2017 and permission to conduct the research from the University Hospital Centre Direction of Libreville. We reported a retrospective study of medical records and archived samples, so all data were fully anonymized before they were accessed and ethics committee waived the requirement for informed consent.

## Results

During the study period, 42,728 birth records were collected, with a prevalence of primary postpartum haemorrhage of 1.6% (n = 671).

### Sociodemographic characteristics of women

The age of parturients ranged from 14 to 46 years with a mean age of 26.9 ± 6.6 years. The 20–39 age group was the most represented, with just over half of the parturients (n = 337). The majority of patients were housewives or women without paid employment (n = 293) and students (n = 156/202). Other characteristics of the women are presented in Table 1.

### Women's gynecological-obstetric history

Table 2 presents the participants' gynaecological and obstetrical histories. Slightly more than half (n = 341/671) of the parturients (50.8%) had no pathological gynecological history at all. In the other half, abortion (n = 301/330) was found in the majority, followed by extrauterine pregnancy and in utero fetal death.

**Table 1. Sociodemographic characteristics of women.**

| Variables | Frequency (n) | Percentage (%) |
|---|---|---|
| Age (years) | | |
| 14–19 | 100 | 14.9 |
| 20–40 | 542 | 80.8 |
| > 40 | 29 | 4.3 |
| Profession | | |
| Household | 302 | 45.0 |
| Student | 202 | 30.1 |
| Public servant | 167 | 24.9 |
| Marital status | | |
| Single | 622 | 92.7 |
| Married | 49 | 7.3 |
| Educational level | | |
| Without education | 01 | 0.1 |
| Primary /secondary | 567 | 84.5 |
| Superior | 103 | 15.4 |

**Table 2. Pregnancy and childbirth characteristics of women at Libreville University Hospital.**

| Variables | Frequency (n) | Percentage (%) |
|---|---|---|
| Surgical history | 671 | |
| Cesarean section | 07 | 1.1 |
| Salpingectomy | 05 | 0.7 |
| None | 659 | 98.2 |
| Gynecological history | 671 | |
| Abortion | 301 | 44.9 |
| Extra uterine pregnancy and other antecedents | 13 | 1.9 |
| No history | 357 | 53.2 |
| Gesture | 671 | |
| 1 | 167 | 24.9 |
| 2–3 | 216 | 32.2 |
| > 3 | 288 | 42.9 |
| Parity | 671 | |
| 1 | 220 | 32.8 |
| 2–3 | 279 | 41.6 |
| > 3 | 172 | 25.6 |
| Gestational age at delivery | 671 | |
| Term | 393 | 58.6 |
| Pre-term | 224 | 33.4 |
| Post term | 54 | 8.0 |
| Types of pregnancy | 671 | |
| Mono fetal | 654 | 97.5 |
| Twin | 17 | 2.5 |
| Starting labour upon arrival | 671 | |
| Immediate | 345 | 51.4 |
| After 3 hours | 326 | 48.6 |
| Duration of labour | 671 | |
| ≤ 12 hours | 636 | 94.8 |
| > 12 hours | 35 | 5.2 |
| Method of delivery | 671 | |
| Caesarean section | 346 | 51.6 |
| Vaginal delivery | 325 | 48.4 |
| Extraction mode | 671 | |
| Without instrument | 664 | 97.8 |
| Forceps | 07 | 2.2 |
| Mode of delivery | 671 | |
| Artificial | 386 | 57.5 |
| Directed | 276 | 41.2 |
| Spontaneous | 9 | 1.3 |
| Birth weight (g)* | 688 | |
| < 2500 | 193 | 28.1 |
| 2500–4000 | 479 | 69.6 |
| > 4000 | 16 | 2.3 |

*: 17 twin pregnancies.

## Characteristics of pregnancy and childbirth

Prenatal follow-up was carried out by 647 parturients (96.4%). Among those who had performed prenatal follow-up and whose records mentioned it, near ¾ (n = 478/647, 73.9%) had performed at least 4 prenatal visits. The remaining quarter (n = 169, 26.1%) had made between 1 and 3 prenatal visits.

Of those who had completed prenatal follow-up, more than three-quarters (n = 498/647, 77.0%) had been followed by a midwife for prenatal visits. Of the remainder, some (n = 139, 21.5%) had been followed by a gynecologist, while the records of the others did not specify the provider of the antenatal visits. Among the records studied, twin pregnancies (n = 17) concerned 2.5% of parturients, while mono-fetal pregnancies (n = 654) concerned 97.5% (Table 2).

The majority of parturients (n = 584/671) had arrived at the University Hospital Centre on their own and by their own efforts, 87%. The remaining patients (n = 8, 13.0%) had been referred from other facilities. More than half of the parturients (n = 345, 51.4%) were not in labour when they were admitted. The other half (n = 326) had been in labour for an average of three hours. Extended work of at least 13 hours was found in 5.2% of parturients (n = 17).

## Etiologies of primary postpartum haemorrhage

The main etiology of this study was haemorrhage of delivery in 65.4% of parturients, the majority of whom had a placental insertion defect (n = 311/439). Genital tract trauma was found as a second etiology (n = 402), with cervical lacerations (n = 176/402) being the main cause. No pathology of haemostasis was identified (Table 3).

## Risk factors for primary postpartum haemorrhage

We observed that primigravidae such as primiparous (OR = 0.20, p < 0.001), multigravidae (OR = 0.32, p < 0.001) and multiparous (OR = 0.48, p = 0.047) women had a lower risk of PPPH than women with 2–3 pregnancies or 2–3 children, this is so even after adjusting with some variables (Adjusted OR = 0.49, p = 0.023 and 0.48, p = 0.012 reespectively). Women with preterm (OR = 1.60, p = 0.005) and post-term (OR = 6.47, p < 0.001) births were more likely to have PPPH than women with full-term births. But after adjusting, only women women with post term births remained to have delivery haemorrhage (Adjusted OR = 2.03, p = 0.042). This

**Table 3. Distribution of parturients by causes of primary postpartum haemorrhage.**

| Etiologies | | Frequency (n) | Percentage (%) |
|---|---|---|---|
| Haemorrhages of delivery# | Placental abruption | 24 | 3.6 |
| | Placenta prævia | 113* | 16.8 |
| | Retroplacental hematoma | 199* | 29.7 |
| | Uterine atony | 103 | 15.4 |
| Trauma of the genital tract# | Cervical laceration | 176 | 26.2 |
| | Vaginal laceration | 96 | 14.3 |
| | Perineal laceration | 44 | 6.6 |
| | Vulvar laceration | 4 | 0.6 |
| | Uterine rupture | 28 | 4.2 |
| | Episiotomy | 54 | 8.0 |
| Coagulation disorder | | 0 | 0 |

*: One parturient presented with both retroplacental hematoma and placenta previa.

#: Some patients presented both some etiologies of delivery haemorrhage and also trauma of the genital tract.

**Table 4. Risk factors for PPPH with etiologies such as delivery haemorrhage in parturients.**

| Variables | | Delivery haemorrhage | | | | | |
|---|---|---|---|---|---|---|---|
| | | Yes | No | OR (95% CI) | p | Adjusted OR* (95% CI) | p |
| Maternal age | | | | | | | |
| | 14–19 | 78 | 22 | 0.83(0.49–1.40) | 0.49 | | |
| | 20–39 | 439 | 103 | 1.00 | - | | |
| | 40 + | 21 | 8 | 0.61(0.26–1.43) | 0.255 | | |
| Gesture | | | | | | | |
| | 1 | 40 | 127 | 0.20(0.13–0.32) | <0.001 | 0.49 (0.29–086) | 0.023 |
| | 2–3 | 131 | 85 | 1.00 | - | 1.0 | - |
| | 4+ | 96 | 192 | 0.32(0.22–0.47) | <0.001 | 0.84 (0.63–0.90) | 0.012 |
| Parity | | | | | | | |
| | 1 | 39 | 181 | 0.01(0.00–0.02) | <0.001 | 0.61(0.48–0.76) | 0.003 |
| | 2–3 | 265 | 14 | 1.00 | - | 1.0 | - |
| | 4+ | 155 | 17 | 0.48(0.23–1.00) | 0.047 | 0.99 (0.72–1.09) | 0.051 |
| Gestationnel age | | | | | | | |
| | Before term | 116 | 108 | 1.60(1.15–2.22) | 0.005 | 1.05(0.32–1.84) | 0.073 |
| | Term | 158 | 235 | 1.00 | - | 1.0 | - |
| | Post term | 74 | 17 | 6.47(3.68–11.38) | <0.001 | 2.03(1.67–2.30) | 0.042 |
| Duration of labour | | | | | | | |
| | ≤ 12 h | 377 | 259 | 1.00 | - | 1.0 | - |
| | 12 h + | 31 | 4 | 5.32(1.86–15.26) | < 0.001 | 1.23(1.02–2.03) | 0.009 |
| Birth weight | | | | | | | |
| | < 2500 g | 100 | 93 | 2.72(1.92–3.86) | < 0.001 | 1.56(1.03–2.01) | 0.007 |
| | 2500–4000 | 357 | 122 | 1.00 | - | 1.0 | - |
| | > 4000 | 4 | 12 | 8.78(2.78–27.73) | < 0.001 | 2.79(1.97–3.01) | 0.006 |
| Mode of delivery | | | | | | | |
| | Artificial | 260 | 126 | 4.13(1.02–16.77) | | 2.21(1.87–3.52) | 0.005 |
| | Directed | 108 | 168 | 1.29(0.31–5.25) | 0.51 | - | - |
| | Spontaneous | 3 | 6 | 1.0 | - | 1.0 | - |

*: Adjusted on surgical history, gynecological history and gestational age at delivery.

was similarly the case for women with children weighing < 2500 g and > at 4000 g. In contrast, women with a duration of labour of more than 12 hours had significantly more PPPH than women with less than 12 hours (Table 4).

Factors associated with genital tract trauma were being 14–19 years of age (Adjusted OR = 2.9, p = 0.008), pauciparum (Adjusted OR = 1.01, p = 0.031), full-term delivery (Adjusted OR = 1.36, p = 0.049) and having a child weighing up 2500 g. Multiparous women were less likely (Adjusted OR = 0.87, p = 0.028) to have genital tract trauma (Table 5).

## Discussion

### Prevalence of PPPH

The prevalence of primary postpartum hemorrhage was 1.6%. This prevalence varies from 0.86% to 9.0% according to studies reported in some countries [5,7,12–18]. In population-based studies, the incidence of PPH is approximately 5% of deliveries when blood loss is not accurately measured and approximately 10% when blood loss is accurately measured [19]. The variation in prevalence for our case during these years could be explained by the fact that not

**Table 5. Risk factors for PPPH with etiologies such as genital tract trauma in parturients.**

| Variables | | Genital tract trauma | | | | | |
|-----------|--|------|------|--------------|--------|----------------------|--------|
| | | Yes | No | OR (95% CI) | p | Adjusted OR¶ (95% CI) | p |
| Maternal age | | | | | | | |
| | 14–19 | 97 | 3 | 11.26 (3.51–36.09) | < 0.001 | 2.9(1.99–3.19) | 0.008 |
| | 20–39 | 402 | 140 | 1.00 | - | 1.00 | - |
| | 40 + | 25 | 4 | 2.18 (0.74–6.36) | 0.145 | - | - |
| Gesture | | | | | | | |
| | 1 | 70 | 97 | 1.00 | - | 1.00 | - |
| | 2–3 | 79 | 137 | 0.80 (0.53–1.21) | 0.287 | - | - |
| | 4+ | 92 | 196 | 0.65 (0.44–0.96) | 0.032 | 1.35 (0.86–2.96) | 0.094 |
| Parity | | | | | | | |
| | 1 | 120 | 100 | 1.00 | - | 1.00 | - |
| | 2–3 | 196 | 83 | 1.97 (1.36–2.85) | <0.001 | 1.01 (1.29–2.00) | 0.031 |
| | 4+ | 20 | 133 | 0.13 (0.07–0.22) | <0.001 | 0.87 (0.67–0.98) | 0.028 |
| Gestationnel age | | | | | | | |
| | Before term | 99 | 125 | 1.00 | - | 1.00 | - |
| | Term | 248 | 145 | 2.16(1.55–3.02) | <0.001 | 1.36(1.11–2.02) | 0.049 |
| | Post term | 35 | 56 | 0.79(0.48–1.29) | 0.350 | - | - |
| Duration of labour | | | | | | | |
| | ≤ 12 h | 469 | 167 | 1.28(0.62–2.68) | 0.555 | - | - |
| | 12 h + | 24 | 11 | 1.00 | - | 1.00 | - |
| Birth weight | | | | | | | |
| | < 2500 g | 96 | 97 | 1.00 | - | 1.00 | - |
| | 2500–4000 | 389 | 90 | 4.37(3.04–6.28) | < 0.001 | 1.99(0.64–3.48) | 0.059 |
| | > 4000 | 11 | 5 | 2.22 (0.74–6.64) | 0.143 | 1.56 (1.49–2.66) | 0.037 |
| Mode of delivery | | | | | | | |
| | Artificial | 126 | 260 | 0.24(0.06–0.98) | 0.024 | 1.25(0.69–2.56) | 0.061 |
| | Directed | 168 | 108 | 0.78(0.19–3.17) | 0.38 | - | - |
| | Spontaneous | 6 | 3 | 1.0 | - | 1.0 | - |

¶: Adjusted on surgical history, gynecological history and gestational age at delivery.

all cases of haemorrhage were reported in the registries. This was due to some low-level haemorrhages that went unnoticed and the fact that this notion had not always been reported in the obstetrical records of the patients because of the burden of work. In addition, the census of maternal deaths at the hospital did not begin until 2014.

## Sociodemographic characteristics

The sociodemographic characteristics studied were age, occupation, marital status and educational level of women.

Women between the ages of 20 and 39 were more represented with more than 80% of the workforce. However, there are variations in age frequency noted in Norway, Tunisia, France and Chad [5,13,14,15,20]. In a general review, Deneux-Thenaux noted the same results [19]. The high frequency of PPPH in this age group in our context could be explained by the fact that it corresponds to the period of increased female genital activity and fertility in the subregion. Any woman in the period of genital activity may be affected by primary postpartum hemorrhage. Housewives and high school students were the most affected classes, with 45.0% and 84.5%, respectively. This result could be explained by the fact that unemployment affects women almost

twice as much as men in Gabon: 20 percent compared to 11 percent. The unemployment rate for young people under 30 years of age is 31% [21]. PPPH seems to be more common among the disadvantaged strata because of the inaccessibility of prenatal care and the management of the factors that contribute to it. Since the marital status of women aged 15–49 in Gabon is dominated by single and cohabiting couples [21], the study found that 92.7% of parturients were single.

## Women's gynecological history

The proportion of women with a history of cicatricial uterus was minimal, 1%, in contrast to that reported by Chouaki in the Democratic Congo, 30.4% [15]. However, the proportion of women with a history of abortion was 45.6%. It has been recognized that curettage and cesarean section are causes of placenta previa and placental retention, which are risk factors for PPPH [12,13]. The absence of a history of PPPH or toxemia gravidarum in the records reviewed is to be deplored. In the partograms, there is no entry mentioning these antecedents, which may be the reason for their absence. Nevertheless, Firmin et al. mentioned a significance between the history of PPPH and its occurrence [17]. Partograms also do not show the estimated amount of blood loss. A new method for estimating blood loss should be adopted, as Andrikopoulou has pointed out [22]. We noted a frequency of PPPH in the pauciparum and primiparum with 37.7% and 32.8%, respectively, compared to 25.6% in the multiparum. The same trend has been observed by some authors, where PPPH was more frequent in paucipares [13,18]. Higher frequencies of PPPH in primiparous women have been observed in some studies, while they have also been observed in multiparous women, as shown in studies in Madagascar and Norway [12,13]. These results show that the frequency of PPPH is as high in pauci pares as in primipares. This could be explained by the overuse of uterotonics in these parturients to speed up labour; the use of indigenous oxytocics at home, foetopelvic disproportions or prolonged labour in primiparous women are also incriminated.

## Characteristics of current pregnancy and childbirth

There was no information on the modes and conditions of evacuation of these parturients, factors that may influence maternal and fetal management and prognosis. Pregnancies considered at term were the most observed and accounted for 53.1%, and one-third of deliveries were premature. In contrast, data from studies conducted in Madagascar and France found a higher frequency of full-term pregnancies than ours [14,18]. There is evidence that premature delivery can lead to placental retention complicated by delivery hemorrhage due to a cleavage defect between the placenta and the myometrium [23].

More than half of the parturients were not in labour when admitted. The other half had been in labour for an average of three hours, and 5.2% of parturients had been in labour for at least 13 hours. In the dossiers explored, there was a lack of information concerning the profile of the staff who had taken immediate care of the parturients, as more than ¾ of women had gone directly to the hospital. Rakotozanany et al. showed that late referral and late management of parturients with PPPH are risk factors for maternal death [12]. The majority of parturients had given birth by Caesarean section. In 2.2% of cases, the extraction is performed by forceps, and an artificial delivery is performed in 57.5% of cases. The frequency of these three procedures seems high since they are performed in the only reference hospital in the city that has an adequate technical platform.

## Etiologies and risk factors for primary postpartum haemorrhage

Delivery haemorrhages accounted for 65.5% of PPPH, while genital traumas accounted for 59.9%. These delivery haemorrhages are dominated by retroplacental haematoma, placenta

previa and uterine atony. Some patients presented with one or two selected causes at the same time. These results are almost similar to the results obtained by some authors who have noted that uterine atony is the main cause of PPPH and that genital tract wounds are responsible for approximately 1 in 5 cases of PPPH [19]. They may be associated with a pathology of delivery mainly represented by uterine inertia and placental retention, a uterine or vaginal genital lesion or a pathology of haemostasis [24,25].

**Mode of delivery.** We noted that delivery was artificial for more than half of the PPPH cases, followed by directed delivery (42.2%). These two proportions, in opposition to spontaneous delivery, appear to be higher than those reported by some authors [21,22,26]. In all instances, these modes of delivery are a risk factor for PPPH. For illustration, Beucher noted that operative vaginal delivery significantly increases the risk of anal sphincter injuy compared with spontaneous vaginal delivery [26].

**The delivery haemorrhages.** The main risk factors for PPPH in the most recent population-based studies vary from one author to another [19]. In our study, primigravida and multi-gestations on the one hand and primiparous and multiparous on the other appeared to have a lower risk of developing delivery haemorrhage. While women with preterm and postterm births, those with more than 12 hours of labor and those with a birth weight of less than 2500 grammes and more than 4000 grammes had a higher risk of developing PPPH. Multiparity, a factor contributing to uterine atony, had a non-negligible proportion in our study. Additionally, the use of oxytocics in our environment is very common, which could explain the frequency of uterine atony. In the occurrence of postpartum hemorrhages after vaginal delivery, the role of placenta previa is classic. The haemorrhage can be explained by the difficulties of uteroplacental cleavage and, above all, by the difficulties of spontaneous haemostasis. Additionally, the women's records did not allow us to note a history of PPPH.

**Trauma of the genital tract.** In contrast to delivery bleeding, women aged 14–19 years are more likely to have genital tract trauma. These results are similar to those found in several studies [19]. Indeed, this young age is exposed to lacerations of the cervix and perineum. We have also noted that women who have given birth at term, those who have children with a birth weight between 2500–4000 grammes and more and women who have given birth more than twice are more likely to have genital tract trauma. Nevertheless, in this study, 98.7% of women underwent artificial and assisted delivery, and 51.6% of women underwent caesarean section. These two factors are recognized as soft tissue tearing factors [26].

**Limitations of the study.** During data collection, we were confronted with the inherent limitations of any retrospective study: incomplete anamnestic information; nonexhaustive paraclinical explorations; and missing medical dossiers (poor management of archives). Despite these constraints, we feel that we have determined the epidemiological aspects of primary postpartum hemorrhage at the University Hospital Centre (UHC). This hospital and monocentric survey does not reflect the epidemiological reality at the level of the whole country. However, it is an advocacy tool to improve the completion and maintenance of admission and follow-up records in the delivery room. Active Management of Third Stage of Labour (AMTSL) was not practiced in the University Hospital Centre of Libreville at the period of this study An intervention study documenting the value of AMTSL would have been more appropriate.

## Conclusions

The prevalence of primary postpartum haemorrhage was 1.6%. The parturients with the highest risk of PPPH were young women aged between 20 and 39 years, unmarried and from an unfavourable socioeconomic background as housewives or schoolchildren, with a secondary

school education and having had at least one abortion and given birth at least once. The most common etiology was haemorrhage from delivery due to a placental defect. Trauma to the genital tract was the second etiology. Multiparity remains the most common risk factor. PPPH is still the leading cause of maternal mortality in Gabon and the rest of the world. This is why it is important to improve management by better assessment of blood loss in the primary post-partum period with the use of collection bags, saving time in diagnosis and management, close monitoring of the parturient and systematic delivery. The three aspects of treatment are insep-arable, justifying adapted multidisciplinary care (obstetrician, anaesthetist, resuscitator, biolo-gist): hence, the interest in strengthening the capacities of health providers in terms of EmONC.

## Acknowledgments

The authors would like to thank the Ministry of Health of Gabon.

## Author Contributions

**Conceptualization:** Nathalie Ledaga Ambounda.

**Data curation:** Alain Jepang Kouanang.

**Formal analysis:** Alain Jepang Kouanang.

**Investigation:** Nathalie Ledaga Ambounda, Liela Agnes Okoyi Ossouka, Vicky Noel Simo Tekem, Eliane Okira Ango.

**Methodology:** Sylvain Honore Woromogo, Felicite-Emma Yagata-Moussa, Vicky Noel Simo Tekem, Alain Jepang Kouanang.

**Software:** Sylvain Honore Woromogo, Eliane Okira Ango, Alain Jepang Kouanang.

**Supervision:** Liela Agnes Okoyi Ossouka, Vicky Noel Simo Tekem, Eliane Okira Ango.

**Validation:** Sylvain Honore Woromogo, Liela Agnes Okoyi Ossouka, Vicky Noel Simo Tekem.

**Visualization:** Eliane Okira Ango.

**Writing – original draft:** Sylvain Honore Woromogo.

**Writing – review & editing:** Nathalie Ledaga Ambounda, Felicite-Emma Yagata-Moussa.

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
