## [Decision Letter · Decision Letter 0]

9 Apr 2021

PONE-D-21-01192

Immediate postpartum haemorrhage at the Libreville University Hospital Centre: Epidemiological profile of women

PLOS ONE

Dear Dr.Woromogo,

Thank you for submitting your manuscript to PLOS ONE. After careful consideration, we feel that it has merit but does not fully meet PLOS ONE’s publication criteria as it currently stands. Therefore, we invite you to submit a revised version of the manuscript that addresses the points raised during the review process.

We look forward to receiving your revised manuscript.

Kind regards,

Claudia Marotta

Academic Editor

PLOS ONE

Journal Requirements:

1. Please ensure that your manuscript meets PLOS ONE's style requirements, including those for file naming. The PLOS ONE style templates can be found athttps://journals.plos.org/plosone/s/file?id=wjVg/PLOSOne_formatting_sample_main_body.pdf and https://journals.plos.org/plosone/s/file?id=ba62/PLOSOne_formatting_sample_title_authors_affiliations.pdf

Additional Editor Comments (if provided):

dear authors follow reviewer suggestions to improve your paper

Reviewers' comments:

Reviewer's Responses to Questions

**Comments to the Author**

1. Is the manuscript technically sound, and do the data support the conclusions?

Reviewer #1: Yes

Reviewer #2: No

2. Has the statistical analysis been performed appropriately and rigorously? 

Reviewer #1: Yes

Reviewer #2: No

3. Have the authors made all data underlying the findings in their manuscript fully available?

Reviewer #1: No

Reviewer #2: No

4. Is the manuscript presented in an intelligible fashion and written in standard English?

Reviewer #1: Yes

Reviewer #2: No

5. Review Comments to the Author

Reviewer #1: 1. Summary of Research

In this paper, the authors sought to determine the prevalence and epidemiological profile of immediate postpartum haemorrhage (IPPH) among women who delivered in Libreville University Hospital Centre. The findings from this work revealed prevalence of IPPH to be 1.6% with delivery haemorrhage as the commonest etiology. Multiparous and young women had the greatest risk.

Below are some comments for the authors:

2. Specific Areas

Abstract:

At line 23, the frequency of maternal deaths related to IPPH is expressed in percentage. Do authors mean prevalence, proportion or otherwise. Clarity on this will be helpful.

In relation to risk factors associated with delivery haemorrhages at line 35, authors could consider using birth weight instead of child weight since the later could be misinterpreted to mean something else.

Repetition of “Prevalence of IPPH was 1.6%” at line 33 and 36.

Introduction:

Primary and secondary PPH are used commonly to mean immediate and late PPH respectively in reproductive health. Authors can include the fact that immediate and primary PPH are used interchangeable to avoid confusion among readers.

Methods:

Provision of brief background information on study area or setting, in relation to bed capacity, deliveries etc will be helpful to contextualize the findings.

Results:

Indicating that more than half of the patients had no gynecological history at all at Line 106 must be rephrased or put in the right context of the study. This can be misleading since all women have gynecological history.

Discussion:

Provision of clearer and detail definition of “haemorrhage after delivery” by authors in the write up will be helpful.

Tables:

Authors can consider using ‘frequency’ to replace ‘numbers’ as a caption in tables 1-3. The appropriate use of ‘N’ and ‘n’ here will enhance accurate presentation of the type of data being presented.

3. Additional Comments

Language editing is recommended. Eg Line 35 genital tract is written as genital track, post-term as postterm.

Authors should clarify and appropriately use ‘n’ or ‘N’ in representing the type of data being presented in the entire manuscript, especially the results section. This will avoid ambiguity, since they mean different things.

Reviewer #2: The topic under consideration is very important as it is the leading cause of maternal mortality in low and middle income countries. There are several missing parts which the authors ought to address.

a). There was no classification of cases into booked and unbooked

b). The obstetric practice was not addressed as well as the cadre of operators

c). Was Active Management of Third Stage of Labour (AMTSL) practiced in this hospital?

d). Were there complications or consequences in these category of patients

e). Was there any intervention

f). It would be more appropriate to compare the active cases with a matched control for better understanding

g). An intervention study documenting the value of AMTSL would have been more appropriate.

h). The language and use of words required to be addressed. The authors might need to engage an English speaking writer

6. PLOS authors have the option to publish the peer review history of their article (what does this mean?). If published, this will include your full peer review and any attached files.

Reviewer #1: No

Reviewer #2: **Yes: **Isaac Folorunso Adewole

---

## [Author Response · Author response to Decision Letter 0]

20 Apr 2021

Response to Reviewers

Editor Comments

1. We ensured that our manuscript meets PLOS ONE's style requirements, including those for file naming.

2. Additional details regarding participant consent were provided in the Methods and online submission information. 

Reviewer #1

Thank you very much for the importance given to our manuscript. We find your comments very pertinent. We have taken these remarks into consideration and corrections have been made in the text.

Abstract 

In fact we mean the proportion of maternal deaths related to IPPH. Using “birth weight˝ instead of child weight in the text. Repetition deleted

Introduction 

We changed terms : using “primary˝ instead of “immediate˝ in the text 

Methods 

Brief background information on study area or setting, in relation to bed capacity, deliveries etc provided.

Results 

We are talking here about a pathological gynecological history. Correction made

Discussion 

We are talking here about delivery heamorrhage.

Tables 

Correction made for Tables 1-3

Additional comments

Correction made. We used ‘tract’ insted of ‘track’ ; ‘post-term’ instead of ‘postterm’ through the manuscript. To avoid ambiguity, ‘N’= birth records (42,728) or study population and ‘n’= number of each item or variable.

Reviewer #2

Thank you very much for the importance given to our manuscript. We find your comments very pertinent. We have taken these remarks into consideration and corrections have been made in the text.

a) In our context we did not do a case control study

b) The obstetric practice was addressed as well as the cadre of operators but this is reported in another document. In fact during the study period, 42,728 birth records were collected, with an incidence of primary postpartum hemorrhage of 1.6% (n = 671). A total of 688 children, including 172 deaths, were born from mothers with PPPH. 

c) Active Management of Third Stage of Labour (AMTSL) was not practiced in the University Hospital Centre of Libreville at the period of this study but is now practised. This could be a limitation of this study. 

d) This question is related to the one asked in point C

e) This question is related to the one asked in point C

f) As suggested, it would be more appropriate to compare the active cases with a matched control for better understanding. We will consider this as a limitation of the study

g) This question is related to the one asked in point C

h) An English speaking writer helped us to verify the language and use of certain words in the text

---

## [Decision Letter · Decision Letter 1]

8 Jun 2021

PONE-D-21-01192R1

Primary postpartum haemorrhage at the Libreville University Hospital Centre : Epidemiological profile of women

PLOS ONE

Dear Dr. WOROMOGO,

Thank you for submitting your manuscript to PLOS ONE. After careful consideration, we feel that it has merit but does not fully meet PLOS ONE’s publication criteria as it currently stands. Therefore, we invite you to submit a revised version of the manuscript that addresses the points raised during the review process.

ACADEMIC EDITOR:

There remain some minor comments from one of the reviewers. In addition please remove reference to qualitative data since it appears no qualitative data was collected. I suggest consulting a statistician to discuss adjusting for potential confounders in your model in table 4. This is an unadjusted model that is presented. I would also suggest adding mode of delivery to your analysis of risk factors for heamorrhage. You data analysis section would need to be updated based on these changes. 

We look forward to receiving your revised manuscript.

Kind regards,

Tanya Doherty, PhD

Academic Editor

PLOS ONE

Journal Requirements:

Reviewers' comments:

Reviewer's Responses to Questions

**Comments to the Author**

1. If the authors have adequately addressed your comments raised in a previous round of review and you feel that this manuscript is now acceptable for publication, you may indicate that here to bypass the “Comments to the Author” section, enter your conflict of interest statement in the “Confidential to Editor” section, and submit your "Accept" recommendation.

Reviewer #1: (No Response)

Reviewer #2: (No Response)

2. Is the manuscript technically sound, and do the data support the conclusions?

Reviewer #1: Yes

Reviewer #2: Yes

3. Has the statistical analysis been performed appropriately and rigorously? 

Reviewer #1: Yes

Reviewer #2: Yes

4. Have the authors made all data underlying the findings in their manuscript fully available?

Reviewer #1: No

Reviewer #2: Yes

5. Is the manuscript presented in an intelligible fashion and written in standard English?

Reviewer #1: Yes

Reviewer #2: Yes

6. Review Comments to the Author

Reviewer #1: I commend the authors for the revision of the manuscript so far.

Below are some few comments though:

Authors did not provide clearer and detail definition of “haemorrhage after delivery or delivery haemorrhage” consistent with literature in the write up. This could have been helpful though.

Again, line 221-222 suggests percentages of delivery haemorrhage and genital trauma as 65.5% and 59.9% respectively, thus exceeding 100%, ie 125% as illustrated in Table 3 (cumulative percentage amounts to 125% and not 100% as indicated). Authors should kindly consider the analysis that yielded such an outcome and correct it appropriately.

Reviewer #2: The revised manuscript is much improved and I recommend acceptance. The limitations of the study should be inserted as a sub-heading.

7. PLOS authors have the option to publish the peer review history of their article (what does this mean?). If published, this will include your full peer review and any attached files.

Reviewer #1: No

Reviewer #2: **Yes: **Professor Isaac Adewole

---

## [Author Response · Author response to Decision Letter 1]

18 Jul 2021

Response to Reviewers

Editor Comments

No qualitative data was collected. We have discussed here qualitative variables such as profession, marital status. There is no reference with qualitative data. 

Finally we used multiple logistic regression taking into account potential confounding factors. Thus, we have Tables 4 and 5

Mode of delivery added in discussion of risk factors for heamorrhage.

The data analysis section updated based on these changes. 

Reviewer #1

Thank you very much for the importance given to our manuscript. 

We defined PPPH in the first sentence of the introduction. We categorised these haemorrhages into 2 groups. The group of haemorrhages due to placental anomalies including uterine atony and the group of haemorrhages of traumatic causes of the genital tract.

After checking our database, some patients presented at the same time with some etiologies of delivery haemorrhage and also trauma of the genital tract. Therefore the total percentage line should be removed to reflect the reality.

Reviewer #2

Thank you very much for the importance given to our manuscript. 

The limitations of the study inserted as a sub-heading.

---

## [Editor Report · Decision Letter 2]

27 Jul 2021

PONE-D-21-01192R2

Primary postpartum haemorrhage at the Libreville University Hospital Centre : Epidemiological profile of women

PLOS ONE

Dear Dr. Woromogo,

Thank you for submitting your manuscript to PLOS ONE. After careful consideration, we feel that it has merit but does not fully meet PLOS ONE’s publication criteria as it currently stands. Therefore, we invite you to submit a revised version of the manuscript that addresses the points raised during the review process.

ACADEMIC EDITOR:

Thank you for this revised manuscript. There are some remaining issues to be addressed:

You still refer to qualitative variables on pages 2 and 4. It would be better to use the terms sociodemographic variables since this is how they are described in table 1. Please move the study limitations section to before the conclusion. In your methods section please provide a definition for artificial, directed and spontaneous mode of delivery. Please have the paper reviewed for English grammar errors.  

We look forward to receiving your revised manuscript.

Kind regards,

Tanya Doherty, PhD

Academic Editor

PLOS ONE
---

## [Author Response · Author response to Decision Letter 2]

3 Sep 2021

Editor Comments

Sociodemographic variables used instead of Qualitative variables in page 2 and 4. 

Limitations of the study moved before Conclusions.

Définition of « artificial, spontatenous and directed delivery » provided in methods section

Paper reviewed for English grammar errors.

---

## [Editor Report · Decision Letter 3]

7 Sep 2021

Primary postpartum haemorrhage at the Libreville University Hospital Centre : Epidemiological profile of women

PONE-D-21-01192R3

Dear Dr. Woromogo,

We’re pleased to inform you that your manuscript has been judged scientifically suitable for publication and will be formally accepted for publication once it meets all outstanding technical requirements.

Kind regards,

Tanya Doherty, PhD

Academic Editor

PLOS ONE
---

## [Editor Report · Acceptance letter]

10 Sep 2021

PONE-D-21-01192R3 

Primary postpartum haemorrhage at the Libreville University Hospital Centre : Epidemiological profile of women 

Dear Dr. Woromogo:

I'm pleased to inform you that your manuscript has been deemed suitable for publication in PLOS ONE. Congratulations! Your manuscript is now with our production department. 

Kind regards, 

on behalf of

Professor Tanya Doherty 

Academic Editor

PLOS ONE